# Effect of Sandblasting Parameters and the Type and Hardness of the Material on the Number of Embedded Al_2_O_3_ Grains

**DOI:** 10.3390/ma16134783

**Published:** 2023-07-02

**Authors:** Beata Śmielak, Leszek Klimek, Kamil Krześniak

**Affiliations:** 1Department of Dental Prosthodontics, Medical University of Lodz, ul. Pomorska 251, 92-213 Lodz, Poland; 2Institute of Materials Engineering, Lodz University of Technology, ul. Stefanowskiego 1/15, 90-924 Lodz, Poland; leszek.klimek@p.lodz.pl; 3Pomeranian Medical University, ul. Rybacka 1, 70-204 Szczecin, Poland; kamil.krzesniak@gmail.com

**Keywords:** abrasive blasting, Cr/Co alloy, Ni/Cr alloy, Ti, ZrO_2_

## Abstract

Background: Is abrasive blasting accompanied by the phenomenon of driving abrasive particles into the conditioned material? Methods: Three hundred and fifteen cylindrical disks of three types of metal alloy (chromium/cobalt, chromium/nickel, titanium, and sintered zirconium dioxide) were divided into four groups (n = 35) and sandblasted at pressures of 0.2, 0.4, or 0.6 MPa with aluminum oxide (Al_2_O_3_), grain size 50, 110, or 250 μm. Then, the surface topography was examined using a scanning microscope, and the amount of embedded grain was measured using quantitative metallography. For each group, five samples were randomly selected and subjected to Vickers hardness testing. In the statistical analyses, a three-factor analysis of variance was carried out, considering the type of material, the size of gradation of the abrasive, and the amount of pressure. Results: The smallest amounts of embedded abrasive (2.62) were observed in the ZrO_2_ treatment, and the largest (38.19) occurred in the treatment of the Ti alloy. An increase in the gradation and the pressure were a systematic increase in the amount of embedded grain. Conclusions: After abrasive blasting, abrasive particles were found on the surface of the materials. The amount of driven abrasive depends on the hardness of the processed material.

## 1. Introduction

Abrasive blasting involves cutting with loose abrasive grain at high speed that generates kinetic energy sufficient to cut. Compressed air, water under pressure (hydro-abrasive treatment), steam, or another medium can be used to set the abrasive grains in motion [1]. Abrasive grains with sharp edges, hitting the treated surface, transfer their impulse completely or partially to a specific area [2,3]. The course of treatment depends on the geometric characteristics of the stream, its energy, and its composition [4]. As a result of abrasive blasting, the obtained surface of the processed material with a different texture. The geometric structure of the surface obtained after machining depends on the variable parameters of the machining process, which include the type of abrasive, the shape and size of abrasive grains, the pressure of the working medium, and the angle of the abrasive stream hitting the surface. The distance of the nozzle from the sandblasted surface can also be a variable. Individual variable process parameters affect the diversification of the surface condition [2,3,4].

The roughness of the surface increases the adhesion of various types of coatings and, at the same time, increases the strength of the connection with the substrate [5,6]. This is a routine method of cleaning castings made from metal alloys; it allows the creation of mechanical “hooks” for anchoring applied fired ceramic masses. This procedure increases the strength of the bond with the veneering ceramic by developing the surface and increasing the contact area between the phases [2,4]. Properly produced rough surfaces can support the distribution of stresses by increasing the dissipation of energy during the action of breaking forces on the connection of materials [7]. During the machining of metal alloys, weakly attached overhangs and alloy flakes produced in the grinding process are also removed, which ensures better surface wettability and better anchoring of deposited coatings [8,9]. In addition, in the process of abrasive blasting, we obtain a comparable, uniform surface condition, which is also important for the creation of a more durable connection of materials [10,11,12].

Ceramic forms a stronger bond with Co/Cr and Ni/Cr alloys than with titanium. Many studies have confirmed a two- or three-fold difference in the strength of such connections with Co/Cr or Ni/Cr alloys compared to titanium alloys [13,14,15,16]. Other studies have found that the adhesion force of porcelain to titanium alloy was 47–64% of that for Cr/Ni alloy—ceramics [17,18]. However, it should be emphasized that the surface roughness played a major role by creating a better anchorage of the ceramics to the metal substrate. Derand and Herø [13] observed that the use of 250 μm alumina abrasive blasting significantly improved the bonding strength of the ceramic to the titanium alloy compared to 50 μm, which may suggest that the embedded particles have a positive effect. It should be emphasized that this is the only way to increase the strength of the connection in the titanium-ceramic system due to the properties of titanium. Yamada et al. [15] examined the adhesion of ceramics to titanium and revealed that abrasive blasting is a very effective method of surface preparation. The adhesion of ceramics to titanium was almost twice as high as compared to that observed in polished samples [19,20].

However, in the case of some metal alloys, such as stainless steel, copper, or titanium alloys, increasing roughness accelerates corrosion [21,22,23]. In addition, during abrasive blasting, abrasive particles with high kinetic energy can stick to the workpiece, which can lead to contamination of the surface [14,24]. Derand and Herø [13] found that Al_2_O_3_ particles could invade up to 10 μm into the treated samples. The contaminated area may hence have less available surface area for connection with ceramics. Gilbert et al. [12] showed that contamination of the titanium surface weakens the mechanical anchoring of dental ceramics, reduces corrosion resistance, and deteriorates the biocompatibility of the material. Additionally, impurities change the topography of the surface, creating a structural discontinuity, which may result in the formation of cracks in the veneering porcelain [25].

Driving abrasive grains with high kinetic energy into the treated surface of prosthetic materials has great practical consequences. Undoubtedly, the particles left in the material contaminate the surface of the treated substrate and reduce the smoothness of the surface [12,21]. With regard to porcelain firing on machined surfaces, the role of embedded particles is not entirely clear. On the one hand, they develop the treated surface, which can improve the quality of the connection; on the other, they can be places where cracks in ceramics can occur [7,21]. There is a strong likelihood that embedded abrasive grains will react with fired ceramics, which may result in the formation of cracks in the veneering porcelain [21]. For example, abrasive blasting may affect the mechanical properties of zirconium oxide [26,27]. Too aggressive action may cause an unfavorable transformation from the tetragonal phase to the undesirable monoclinic (t → m) [24,25,26,27,28,29]. As the transformation phase increases from the surface of the sample to the entire volume of the sample, microcracks, and residual stresses may develop and decrease the bending force [26,27].

Therefore, it is important to determine the number of embedded grains for individual processing parameters. Despite many studies, the combination of ceramics with zirconium oxide and ceramics with titanium alloys is still the weakest point of prosthetic restorations and leads to chipping and fractures.

The aim of the work is to examine the effect of abrasive blasting on the amount of Al_2_O_3_ abrasive driven into the surface of Cr/Co, Cr/Ni, and Ti alloys and synthesized ZrO_2_, as well as to find a relationship between the amount of driven abrasive and the hardness of the materials.

## 2. Materials and Methods

Three hundred and fifteen cylindrical disks with a diameter of 9 mm and a height of 5 mm from three types of metal alloys: Cr/Co (Heraenium^®^ P, Heraeus Holding GmbH, Hanau, Germany), Ni/Cr (Wiron 99, BEGO USA Inc., Lincoln, RI, USA), Ti (Tritan CpTi 1, DENTAURUM GmbH & Co. KG, Ispringen, Germany), and 315 of the same dimensions of sintered ZrO_2_ 3TPZ-Y (Cermill, Amann Girrbach AG, Koblach, Austria) were used in the study. The samples were divided into groups (n = 35), and subjected to abrasive blasting with aluminum oxide (Al_2_O_3_) with grain sizes 50, 110, and 250 μm at pressures of 0.2, 0.4, or 0.6 MPa. To unify the surface of the samples before the test, each surface was ground with SiC sandpapers with grains of 220, 400, 600, and 800, respectively, on a Metasinex grinder with water cooling. Then, the samples were washed in an ultrasonic cleaner (Quantrex 90 WT, L&R Manufacturing, Inc., Kearny, NJ, USA) in ethyl alcohol for 10 min and dried with compressed air. Abrasive blasting was carried out on the Mikroblast Duo device (Prodento - Optimed, Warsaw, Poland). The working distance from the nozzle was 20 mm, the angle of incidence of the abrasive was 45°, and the treatment time was 20 s.

The samples prepared this way were observed in a scanning electron microscope (SEM, HITACHI S3000-N, Hitachi, Ltd., Tokyo, Japan). The surface topography of the samples was obtained with the use of secondary electrons (SE) and the so-called material contrast in the light of backscattered electrons (BSE). Observations conducted in this way, due to the material contrast resulting from the difference in chemical composition, make it possible to determine the areas occupied by abrasive material grains stuck into the surface, which was confirmed in works [12,30,31]. Example photos of samples of metal alloys and zirconium dioxide after abrasive blasting obtained using backscattered electrons BSE are shown in Figure 1. Ten photos were taken at randomly-selected locations of each sample. The surface fraction of the abrasive material particles was determined by quantitative metallography with Mentilo software [22]. From each group, five samples were subjected to Vickers hardness measurements on a KB Prüftechnik hardness tester after a load of 9.81 N (1 kG).

Dark areas appeared on the surface of metal alloy samples after abrasive blasting, indicating a difference in the chemical composition compared to the treated substrate. Quantitative metallography methods with the application of the Metillo program were used to determine the surface fraction of abrasive particles [22,27]. Briefly, the microscope image was loaded into Metillo and subjected to the following procedures: shadow correction, normalisation of the grey-level histogram, and manual binarization. The percentage share of the dark (red) areas of the total abrasive elements embedded in the surface of the sample was then calculated. Figure 2a shows an exemplary microscopic photo presenting the manual binarization of the image of a ZrO_2_ sample after abrasive blasting with Al_2_O_3_ grain size 110 μm and pressure 0.2 MPa. The red areas in Figure 2b are aluminum particles embedded in the surface of the sample.

Statistical analyses of the results were performed using the PQStat statistical package, version 1.8.2.218. A three-factor analysis of variance was carried out comprising the type of material, the size of the gradation of the abrasive, and the amount of pressure. Tukey’s test was performed as a post hoc test. Test probability was considered significant for *p* < 0.05 and highly significant for *p* < 0.01.

## 3. Results

The amount of driven Al_2_O_3_ abrasive used for abrasive blasting of four different materials with variable processing parameters is presented in Table 1. The graphical interpretation of the results is shown in Figure 3. The smallest amounts of driven abrasive were observed for ZrO_2_ treatment, and among them, the smallest was noted for a value of 250 μm of the abrasive gradation and pressure of 0.6 MPa. In the case of such a combination of factors, the mean value (2.62) of results of the driven abrasive did not differ significantly (*p* > 0.05) from the combination of factors with a pressure of 0.4 MPa (2.88) and 0.2 MPa (3.21). In other words, for ZrO_2_ treatment with an abrasive gradation of 250 μm, regardless of the applied pressure, the amount of embedded abrasive was small and constituted a homogeneous group. (Indicated by the letter “a”). Significantly (*p* > 0.05) larger amounts of embedded abrasive were noted for ZrO_2_ machining and grain gradation of 110 μm or 50 μm. Then, regardless of the applied pressure, the average amounts of driven abrasive made up another homogeneous group (marked with the letter “b”). All other combinations of factors resulted in significantly (*p* > 0.05) more embedded abrasive than those found for ZrO_2_ treatment. Among all the results, the largest amount of grain was embedded in the treatment of the Ti alloy at a pressure of 0.6 MPa and grain gradation of 250 μm. The average amount of embedded grain was 38.19 and was significantly (*p* > 0.05) higher than all other treatment variants.

Descriptive statistics of the amount of driven Al_2_O_3_ abrasive depending on the main factors and the analysis of the variance table are presented in Table 2. All interactions between the factors were highly significant (*p* < 0.0001). In general, a highly significant (*p* < 0.0001) difference in the amount of embedded abrasive depending on the material subjected to treatment was noted. Highly significant differences (*p* < 0.0001) were observed in each material which was compared with any other material. The smallest amount of embedded abrasive was noted for ZrO_2,_ and the highest amount—for the Ti alloy.

A comparison of gradations shows a highly significant difference (*p* < 0.0001) between them. Highly significant differences (*p* < 0.0001) were observed for each gradation which was compared to any other gradation. The lowest amounts of embedded abrasive were observed for 50 μm gradation, and the highest amounts of embedded abrasive were noted at a gradation of 250 μm. This means that the increase in the gradation of the abrasive systematically increased the amount of embedded grain.

A pressure comparison indicated a highly significant difference (*p* < 0.0001) between them. Highly significant differences (*p* < 0.0001) were observed for each pressure compared to any other pressure. The smallest amounts of driven abrasive were observed for a pressure of 0.2 MPa, and the largest amounts of driven abrasive were noted for a pressure of 0.6 MPa. In the case of the metal alloys Ti, Ni/Cr, and Co/Cr, an increase in pressure was associated with a higher number of particles knocked out; however, no such effect was observed for ZrO_2._

All factor interactions were highly significant (*p* < 0.0001). In general, a highly significant (*p* < 0.0001) difference in the amount of embedded abrasive depending on the material was found. Highly significant differences (*p* < 0.0001) were observed in each material which was compared with any other material. The smallest amount of driven abrasive was observed during the machining of ZrO_2,_ and the largest—during the machining of the Ti alloy. Comparison of abrasive gradations showed highly significant differences (*p* < 0.0001) between them. A highly significant difference (*p* < 0.0001) was observed for all gradations, which were compared to any other gradations. The smallest amount of embedded abrasive was at 50 μm gradation, and the largest was at 250 μm gradation. This means that the increase in the gradation of the abrasive systematically increased the number of embedded grains. A comparison of pressure values indicated a highly significant difference (*p* < 0.0001) between them. Highly significant differences (*p* < 0.0001) were observed when each pressure was compared to any other pressure. The smallest amounts of driven abrasive were noted for a pressure of 0.2 MPa, and the largest amounts for a pressure of 0.6 MPa. This means that the increase in the pressure systematically increased the amount of embedded abrasive. Results of hardness measurements using the Vickers HV1 method of individual groups of samples of four different materials used for testing at a load of 9.81 N (1 kG) are presented in Table 3.

ZrO_2_ was found to be the hardest substance (mean 1432 HV) and Ti alloy the least (mean 96 HV), with ZrO_2_ demonstrating about 15 times greater hardness than the Ti alloy. The mean hardness values were 185 HV for the Ni/Cr alloy and 403 HV for Co/Cr.

## 4. Discussion

Based on microscopic observations (in backscattered electrons, i.e., material contrast) of the surface of the samples after abrasive blasting, components not belonging to the tested materials were found. Based on previous works, it can be concluded that these are grains of abrasive material embedded in the surfaces of the samples [12,30,31].

When analyzing the phenomena occurring between the grains of the abrasive material and the machined surface, three processes should be considered: grains perform cutting work and bounce off the surface, grains perform cutting work and remain fixed in the surface, or grains stick to the surface without cutting. The grains of the cutting material carried by compressed air carry some energy depending on their mass and speed, which is the result of applied air pressure. The quality of the phenomenon that will occur depends on the energy of the falling grain in relation to the energy needed to perform the cutting work, as well as on its orientation at the moment of contact with the machined surface. It should be emphasized that the abrasive grains are irregular polygons. Therefore, when considering the influence of machining parameters, the type of abrasive grain, its gradation, the applied pressure, and the hardness of the workpiece should be considered.

The amount of embedded abrasive particles was found to depend on the type of material, the abrasive gradation, and the amount of applied pressure. The tested materials were characterized by different hardness, from very soft titanium (97 HV) through a slightly harder Ni/Cr alloy (195 HV), harder Cr/Co alloy (300 HV), to very hard ZrO_2_ (1300 HV). It should be emphasized that the hardness of ZrO_2_ was close to the hardness of the used abrasive materials. The largest amount of driven abrasive, regardless of the type of abrasive material and processing parameters, was observed on the surfaces of samples made of titanium alloy, followed by Ni/Cr and Co/Cr alloys, and the smallest amount was noted for the surfaces of ZrO_2_ samples. Here we can see a clear and most important dependence of the number of driven grains on the hardness of the workpiece—the higher its hardness, the fewer stuck grains with the same machining parameters. The surface occupied by embedded particles can increase many times over. Depending on the parameters used, it ranges from 2.62 to 7.58% for ZrO_2_, through 13.51 to 25.50% for the Cr/Co alloy, 17.63 to 30.48% for the Ni/Cr alloy, up to 23.64 to 38.19% for the Ti alloy. This relationship can be explained by the relationship between the grain energy and the energy needed to perform cutting work. We can observe a complete loss of energy by a particle that is stopped at the moment of contact with the treated surface or a partial loss of energy during which the particle bounces off the treated surface. As ZrO_2_ is harder than aluminum oxide, fewer particles become embedded. As such, the differences are not as visible. However, it is not obvious how many grains will be driven in, and this has a direct impact on the bond between the framework and the veneering ceramics.

The obtained results are for practical use. As for other parameters, there may also be sandblasting time and particle incidence angle. However, they are of no practical importance. It was found that after several seconds of sandblasting, the condition of the surface did not change. On the other hand, the angle of incidence of the abrasive has an undoubted effect on the amount driven into the surface. However, in prosthetic practice, due to the shape of the processed elements, it is not possible to maintain a constant angle; therefore, we did not consider this parameter.

The size of the abrasive grains used for processing is another parameter influencing the number of embedded particles. The increase in the gradation of the abrasive systematically increased the amount of embedded grain. The change in the size of the abrasive grain was associated with an increase in its weight and resistance to crushing, and thus a change in cutting properties, i.e., also in the geometric structure of the machined surface. Smaller abrasive grains have lower kinetic energy due to their lower mass. The angles of the tops of the abrasive grains also change by changing their geometric dimensions [1,3,4]. It seems that, for example, assuming a similar number of embedded grains, a five-fold increase in the grain size from 50 m to 250 μm should cause a large difference in the area they occupy. However, this did not happen because the accelerated grain cracked and crumbled into smaller parts after its contact with the treated surface. Thus, a fragment of the grain and not the whole of it was driven in, which resulted in a smaller number of grains than expected.

The dependence of the number of embedded grains on the treatment pressure can be explained in a similar way. Increased pressure systematically increased the amount of driven abrasive. However, these relationships were not as clear as in the case of the type and hardness of the material. Increased pressure resulted in greater kinetic energy of the grains and thus also a more intense cutting process [2,3,4,7]. This will have an undeniable effect on improving the mechanical anchoring and increasing the wettability of the treated surface [6,8,17,27,32]. However, a more accurate explanation of the role of embedded abrasive particles requires further research.

Referring to many studies, it can be stated that the abrasive blasting process has a beneficial effect on most surfaces of materials prepared for bonding with other materials. Additionally, this process is necessary to obtain a surface characterized by parameters that will create a mechanically durable connection. The surface roughness obtained in this way plays a major role in improving the quality of the connection, which can be better anchored in the surface layer of the base material of particles of the applied coating.

While the obtained results have practical significance, sandblasting time and particle incidence angle do not, despite playing a role in embedding. It was found that several seconds of sandblasting did not appear to influence the condition of the surface.

Although the angle of incidence of the abrasive has an undoubted effect on embedding, it is not possible to maintain a constant angle in prosthetic practice due to the shape of the processed elements. Therefore, this parameter was not included in the analysis.

Although sandblasting hardens the surface of the metal, this has no practical significance due to the hardening being negated by the recrystallization that occurs when the ceramic is fired at 900 °C.

## 5. Conclusions

After abrasive blasting, abrasive particles were found on the surface of treated materials. The amount of driven abrasive depends on the hardness of the processed material, the gradation of the abrasive, as well as the size of the working pressure used during machining. The greater the hardness of the processed material, the smaller the number of grains driven into its surface. In the case of metal alloys: Ti, Ni/Cr, and Co/Cr, an increase in pressure was correlated with an increase in the number of particles knocked out; however, this effect was not observed for ZrO_2._ The aim of the work, however, was not to emphasize the advantages of sandblasting but to draw attention to its effect on grain embedding, which may affect any subsequent bond formed with ceramics.

## Figures and Tables

**Figure 1 materials-16-04783-f001:**
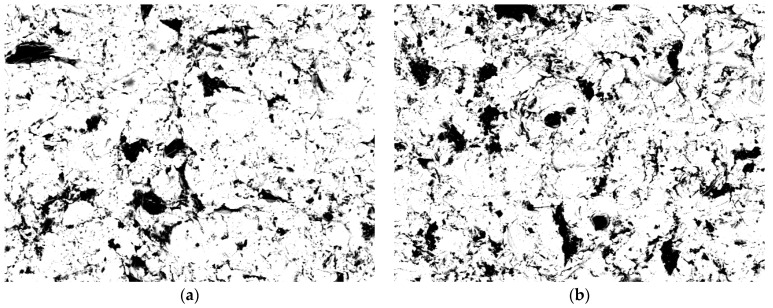
Images obtained using backscattered electrons BSE after abrasion with Al_2_O_3_ 250 μm particles and under the pressure of 0.2 MPa with (magnification 500×): (**a**) Cr/Co alloy, (**b**) Ni/Cr alloy, (**c**) Ti alloy, and (**d**) ZrO_2_.

**Figure 2 materials-16-04783-f002:**
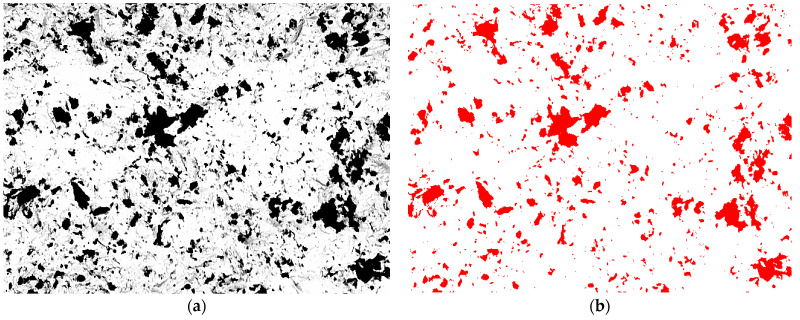
Binarization of the microscopic image: (**a**) initial image and (**b**) image after binarization (for calculations).

**Figure 3 materials-16-04783-f003:**
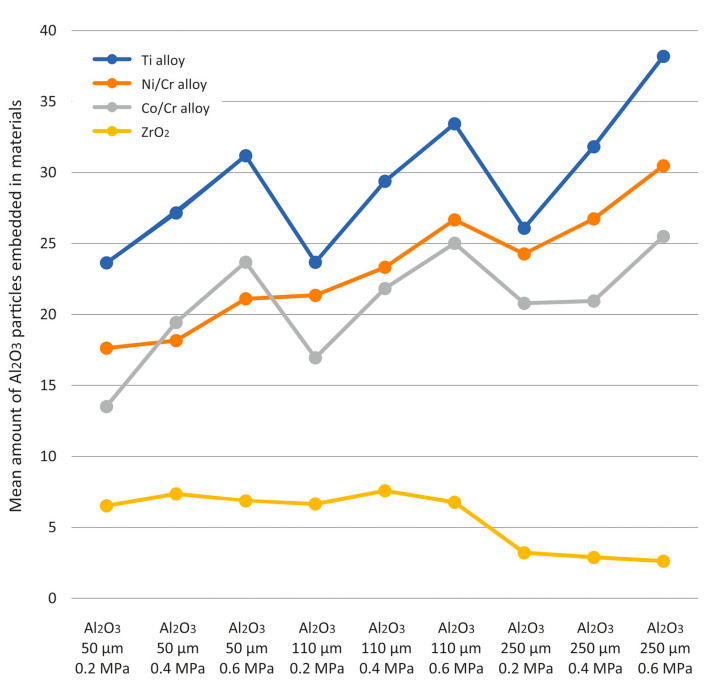
The amount of Al_2_O_3_ particles embedded in the four different test materials according to abrasion parameters.

**Table 1 materials-16-04783-t001:** Descriptive statistics of the amount of Al_2_O_3_ embedded abrasive according to material and airborne-particle abrasion parameters.

Material	Gradation (μm)	Pressure (MPa)	Arithmetic Mean	Standard Deviation	Standard Error of the Mean	Uniform Groups (Tukeys Post Hoc Test)
alloy Ti	50	0.2	23.64	1.71	0.29	Ij
		0.4	27.16	1.91	0.32	n
		0.6	31.19	1.88	0.32	p
	110	0.2	23.68	1.76	0.30	ij
		0.4	29.39	1.52	0.26	o
		0.6	33.44	2.06	0.35	r
	250	0.2	26.08	1.43	0.24	lmn
		0.4	31.82	1.91	0.32	p
		0.6	38.19	1.82	0.31	s
alloy Ni/Cr	50	0.2	17.63	4.32	0.73	d
		0.4	18.16	2.00	0.34	de
		0.6	21.11	2.01	0.34	g
	110	0.2	21.35	2.38	0.40	g
		0.4	23.33	1.40	0.24	hi
		0.6	26.67	2.22	0.38	mn
	250	0.2	24.27	1.53	0.26	ijk
		0.4	26.74	1.53	0.26	mn
		0.6	30.48	2.32	0.39	op
alloy Co/Cr	50	0.2	13.51	1.50	0.25	c
		0.4	19.44	2.02	0.34	ef
		0.6	23.69	1.54	0.26	ij
	110	0.2	16.95	1.67	0.28	d
		0.4	21.83	1.81	0.31	gh
		0.6	25.02	1.14	0.19	jkl
	250	0.2	20.79	1.65	0.28	fg
		0.4	20.95	1.66	0.28	fg
		0.6	25.50	1.20	0.20	klm
ZrO_2_	50	0.2	6.52	0.44	0.07	b
		0.4	7.36	0.49	0.08	b
		0.6	6.87	0.35	0.06	b
	110	0.2	6.65	0.45	0.08	b
		0.4	7.58	0.37	0.06	b
		0.6	6.77	0.38	0.06	b
	250	0.2	3.21	0.45	0.08	a
		0.4	2.88	0.40	0.07	a
		0.6	2.62	0.39	0.07	a

Means marked with the same letter did not differ significantly from each other (*p* > 0.05), while if there was no common letter between the two compared means, then the means differed significantly (*p* > 0.05).

**Table 2 materials-16-04783-t002:** Descriptive statistics of the amount of driven Al_2_O_3_ abrasive depending on the main factors and the table of the analysis of variance.

		Arithmetic Mean	Standard Deviation	Standard Error of the Mean	Uniform Groups (Tukeys Post Hoc Test)
Material	alloy Ti	29.40	4.87	0.27	d
	alloy Ni/Cr	23.30	4.61	0.26	c
	alloy Co/Cr	20.85	3.97	0.22	b
	ZrO_2_	5.61	1.99	0.11	a
Gradation [μm]	50	18.02	8.01	0.39	a
	110	20.22	8.74	0.43	b
	250	21.13	11.56	0.56	c
Pressure [MPa]	0.2	17.02	7.75	0.38	a
	0.4	19.72	9.04	0.44	b
	0.6	22.63	11.04	0.54	c
		F	*p*	Eta-squared partially	
Material		11,312.09	<0.0001	0.9652	
Gradation [μm]		375.41	<0.0001	0.3802	
Pressure [MPa]		1158.38	<0.0001	0.6543	
Material*Gradation		261.33	<0.0001	0.5616	
Material*Pressure [MPa]		170.07	<0.0001	0.4547	
Gradation*Pressure [MPa]		7.27	<0.0001	0.0232	
Material*Gradation*Pressure [MPa]		18.13	<0.0001	0.1509	

Means marked with the same letter did not differ significantly from each other (*p* > 0.05), while if there was no common letter between the two compared means, then the means differed significantly (*p* > 0.05).

**Table 3 materials-16-04783-t003:** Results of HV1 hardness measurements of the tested materials.

Material			
Alloy Ti	Alloy Ni/Cr	Alloy Co/Cr	ZrO_2_
94	187	405	1410
95	186	400	1405
97	186	396	1410
101	184	422	1399
90	189	393	1432
98	182	402	1416
X mean = 96	X mean = 185	X mean = 403	X mean = 1412
SD = 3.8	SD = 2.4	SD = 10.2	SD = 11.3

## Data Availability

Not applicable.

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
