# Peer review of "Effect of Sandblasting Parameters and the Type and Hardness of the Material on the Number of Embedded Al_2_O_3_ Grains"

_materials, 2023, doi:10.3390/ma16134783_

Round 1

Reviewer 1 Report

This paper demonstrates the abrasive blasting experiment results. Authors chose four kinds of metal alloy materials as substrate and aluminum oxide as sand. They carried out a three-factor analysis of variance to describe the effects of type and hardness of materials and pressure and diameter of sand on the number of embedded Al2O3 grains. This experiment is already complete, however, there are some points below authors may need to check before publication.

(1)   Between line 205 to line 210, authors indicate that “the increase in pressure systematically increased the amount of driven abrasive.” But, ZrO2 is a less significant example. Increasing pressure does not increase the amount of driven abrasive apparently.

(2)   The Discussion part is too simplified. The reason authors proposed are obvious. For example, if the substrate owns the larger hardness, naturally there are less embedded grains. Therefore, it seems more like an experiment report rather than detailed discussion or analysis.

(3)   Also, the Conclusion part is too simplified. Authors just repeat the conclusions found in the experiment.  

(4)   In conclusion, I think this paper indeed states the experiment results detailed. But it lacks necessary analyse.

This paper demonstrates the abrasive blasting experiment results. Authors chose four kinds of metal alloy materials as substrate and aluminum oxide as sand. They carried out a three-factor analysis of variance to describe the effects of type and hardness of materials and pressure and diameter of sand on the number of embedded Al2O3 grains. This experiment is already complete, however, there are some points below authors may need to check before publication.

(1)   Between line 205 to line 210, authors indicate that “the increase in pressure systematically increased the amount of driven abrasive.” But, ZrO2 is a less significant example. Increasing pressure does not increase the amount of driven abrasive apparently.

(2)   The Discussion part is too simplified. The reason authors proposed are obvious. For example, if the substrate owns the larger hardness, naturally there are less embedded grains. Therefore, it seems more like an experiment report rather than detailed discussion or analysis.

(3)   Also, the Conclusion part is too simplified. Authors just repeat the conclusions found in the experiment.  

(4)   In conclusion, I think this paper indeed states the experiment results detailed. But it lacks necessary analyse.

Author Response

This paper demonstrates the abrasive blasting experiment results. Authors chose four kinds of metal alloy materials as substrate and aluminum oxide as sand. They carried out a three-factor analysis of variance to describe the effects of type and hardness of materials and pressure and diameter of sand on the number of embedded Al2O3 grains. This experiment is already complete, however, there are some points below authors may need to check before publication. 

  • Between line 205 to line 210, authors indicate that “the increase in pressure systematically increased the amount of driven abrasive.” But, ZrO2is a less significant example. Increasing pressure does not increase the amount of driven abrasive apparently

Answer: The Analyses of Variance (Table 2, page 8) shows that the material had a highly significant impact. The post hoc analyses clearly indicate that the ZrO2results are the lowest (indicated by letters). In addition, ZrO2 was found to demonstrate little variation compared to other materials (Table 1, page 6). There are only two letters a and b, which is the lowest according to the research results. There is no difference between the pressures within the “a” gradation and gradations 50 and 110 are no different. The same is depicted in graph 3 on page 7. This has been added to the statistics article.

" In the case of the metal alloys Ti, Ni/Cr, Co/Cr, an increase in pressure was associated with a higher number of particles knocked out; however, no such effect was observed for ZrO2.”

  • The Discussion part is too simplified. The reason authors proposed are obvious. For example, if the substrate owns the larger hardness, naturally there are less embedded grains. Therefore, it seems more like an experiment report rather than detailed discussion or analysis. 

Answer: The Discussion section has been supplemented with the following: It is known that the higher the hardness, the less embedded particles. However, it is not obvious how many grains will be driven in, and this has a direct impact on the connection of the framework with the veneering ceramic. The obtained results have practical value ..

“As ZrO2 is harder than aluminum oxide, fewer particles become embedded. As such, the differences are not as visible. However, it is not obvious how many grains will be driven in, and this has a direct impact on the bond between the framework and the veneering ceramics.”

“The obtained results are for practical use. As for other parameters, there may also be sandblasting time and particle While the obtained results have practical significance, sandblasting time and particle incidence angle do not, despite playing a role in embedding. It was found that several seconds of sandblasting did not appear to influence the condition of the surface.

Although the angle of incidence of the abrasive has an undoubted effect on embedding, it is not possible to maintain a constant angle in prosthetic practice due to the shape of the processed elements. Therefore, this parameter was not included in the analysis.

Although sandblasting hardens the surface of the metal, this has no practical significance, due to the hardening being negated by the recrystallization that occurs when the ceramic is fired at 900°C.”

  • Also, the Conclusion part is too simplified. Authors just repeat the conclusions found in the experiment. 

Answer: The conclusions have been supplemented.

“In the case of metal alloys: Ti, Ni/Cr, Co/Cr, an increase in pressure was correlated with an increase in the number of particles knocked out; however, this effect was not observed for ZrO2. The aim of the work, however, was not to emphasize the advantages of sandblasting, but to draw attention to its effect on grain embedding, which may affect any subsequent bond formed with ceramics.”

  • In conclusion, I think this paper indeed states the experiment results detailed. But it lacks necessary analyse. 

Answer: The article stated: “Statistical analyses of the results were performed using the PQStat statistical package, version 1.8.2.218. A three-factor analysis of variance was carried out comprising the type of material, the size of the gradation of the abrasive and the amount of pressure. Tukey’s test was performed as a post hoc test. Test probability was considered significant for P<0.05, and highly significant for P<0.01.”

This analysis is a standard procedure and is adequate for the study. It seems to us that in a serious article you cannot copy the definition of analysis of variance from a statistics textbook or write an essay on what a post-hoc test is for.

Reviewer 2 Report

The paper "Effect of sandblasting parameters and the type and hardness of the material on the number of embedded Al2O3 grains" presents interesting results of experimental research carried out in the field of sandblasting of materials used in the medical field. I believe that the paper could be published in "Materials" after its review by the authors:

- it would be useful to create an experimental plan and follow it;

- the presentation of the sandblasting installation would be useful;

- it would be useful to indicate how to measure the amount of particles incorporated in the processed material.

- it would be useful to explain the statistical analysis;

- it would be interesting to study the influence of other parameters of the sandblasting process on the amount of incorporated particles. The conclusions reached by the authors were predictable, and a mathematical modeling would increase the importance of the obtained results.

Thank you

Author Response

The paper "Effect of sandblasting parameters and the type and hardness of the material on the number of embedded Al2O3 grains" presents interesting results of experimental research carried out in the field of sandblasting of materials used in the medical field. I believe that the paper could be published in "Materials" after its review by the authors: 

- it would be useful to create an experimental plan and follow it;

Answer: Could you be more precise regarding the content of the experimental plan, and what is missing?

- the presentation of the sandblasting installation would be useful;

Answer: One of the sandblasters commonly used in the practice of making prosthetic restorations was used for sandblasting. We considered adding a photo of the equipment but did not feel it to be necessary, as it is a standard sandblaster used for this purpose.

- it would be useful to indicate how to measure the amount of particles incorporated in the processed material.

Answer: This is described in the article in an abbreviated version. The full methodology is described in the publication:

  1. Śmielak, B.; Klimek, L. Effect of air abrasion on the number of particles embedded in zirconia. Materials201811, 259

We did not consider it advisable to describe it again to avoid plagiarism. But as suggested, added to the article:

“Briefly, the microscope image was loaded into Metillo and subjected to the following procedures: shadow correction, normalisation of grey level histogram and manual binarization. The percentage share of the dark (red) areas of the total abrasive elements embedded in the surface of the sample was then calculated.”

- it would be useful to explain the statistical analysis;

Answer: The article stated: “Statistical analyses of the results were performed using the PQStat statistical package, version 1.8.2.218. A three-factor analysis of variance was carried out comprising the type of material, the size of the gradation of the abrasive and the amount of pressure. Tukey’s test was performed as a post hoc test. Test probability was considered significant for P<0.05, and highly significant for P<0.01.”

This analysis is a standard procedure and is adequate for the study. It seems to us that in a serious article you cannot copy the definition of analysis of variance from a statistics textbook or write an essay on what a post-hoc test is for.

- it would be interesting to study the influence of other parameters of the sandblasting process on the amount of incorporated particles. The conclusions reached by the authors were predictable, and a mathematical modeling would increase the importance of the obtained results.

Answer: Due to the extensiveness of the issue, mathematical modeling may be a part of the next article (as it is in position Pietnicki K., Wołowiec E., Klimek L.: Modeling of the number of stubble stuck elements after abrasive jet machining-processing. Archives of Foundry Engineering vol. 11, (3) 2011, p. 51 – 54.

This has been added to the article. “While the obtained results have practical significance, sandblasting time and particle incidence angle do not, despite playing a role in embedding. It was found that several seconds of sandblasting did not appear to influence the condition of the surface. Although the angle of incidence of the abrasive has an undoubted effect on embedding, it is not possible to maintain a constant angle in prosthetic practice due to the shape of the processed elements. Therefore, this parameter was not included in the analysis. Although sandblasting hardens the surface of the metal, this has no practical significance, due to the hardening being negated by the recrystallization that occurs when the ceramic is fired at 900°C.”

Reviewer 3 Report

The authors investigated the amount of embedded abrasive particles after sandblasting process. It was found that after abrasive blasting, abrasive particles were found on the surface of the materials. The amount of driven abrasive depended on the hardness of the processed material. This topic has value to the engineering applications. However, some issues should be addressed before publication.

1.     Introduction. Other surface engineering techniques should be also introduced combining some related literatures, including PEO (https://doi.org/10.1016/j.surfcoat.2023.129426; https://doi.org/10.1016/j.msec.2018.04.057), laser surface texturing (https://doi.org/10.1016/j.carbon.2020.05.041), to highlight the advantages of sandblasting (like low-cost, easy-processing, etc.)

2.     Besides the surface hardness, the roughness of the samples also has influence on the results. Are the samples have similar surface roughness?

3.     Sandblasting process sometimes leads to hardening of the sample surface, which should be discussed in this manuscript.

Author Response

The authors investigated the amount of embedded abrasive particles after sandblasting process. It was found that after abrasive blasting, abrasive particles were found on the surface of the materials. The amount of driven abrasive depended on the hardness of the processed material. This topic has value to the engineering applications. However, some issues should be addressed before publication. 

  1. Other surface engineering techniques should be also introduced combining some related literatures, including PEO (https://doi.org/10.1016/j.surfcoat.2023.129426; https://doi.org/10.1016/j.msec.2018.04.057), laser surface texturing (https://doi.org/10.1016/j.carbon.2020.05.041), to highlight the advantages of sandblasting (like low-cost, easy-processing, etc.) 

Answer: Thank you for these references: these are very interesting items, which we hope to incorporate in our future research. Of course, there are other surface texturing techniques that we have been working on.

This was added in the article:

“The aim of the work, however, was not to emphasize the advantages of sandblasting, but to draw attention to its effect on grain embedding, which may affect any subsequent bond formed with ceramics.”

  1. Besides the surface hardness, the roughness of the samples also has influence on the results. Are the samples have similar surface roughness?  

Answer: Of course, other factors also affect the quality of the connection. But it has been presented in several other works, and we did not consider it necessary to repeat it.

  1. Pietnicki K., Wołowiec E., Klimek L.: The effect of abrasive blasting on the strength of a joint between dental porcelain and metal base” Acta of Bioengineering and Biomechanics Vol. 16, No. 1, 2014, s. 63 – 68,
  2. Golebiowski M., Wolowiec E., Klimek L.: Airborne-particle abrasion parameters on the quality of titanium-ceramic bonds. Journal of Prosthetic Dentistry, Vol. 113, Issue 6, p. 453 – 459 Published online: March 4 2015,
  3. Śmielak B., Gołębiowski M, Klimek L., Wołowiec E.: Effect of Surface Treatment of Titanium Elements on The Bond Strength to Zirconium Dioxide, Solid State Phenomena Vol. 225 (2015) pp 151 – 158,
  4. Śmielak B., Klimek L., Wojciechowski R., Bąkała M.: Effect of  zirconia  surface  treatment  on  its  wettabilityby  liquid  Journal of Prosthetic Dentistry. Volume 122, Issue 4, October 2019, Pages 410.e1-410.e6, DOI: doi.org/10.1016/j.prosdent.2019.06.021,
  5. Czepułkowska W. Korecka-Wołowiec E., Klimek L.: The Condition of Ni-Cr Alloy Surface After Abrasive Blasting with Various Parameters. Journal of Materials Engineering and Performance, httorg/10.1007/s11665-019-04399-z
  6. Czepułkowska-Pawlak W., Wołowiec-Korecka E., Klimek L.: The Surface Condition of Ni-Cr after SiC Abrasive Blasting for Applications in Ceramic Restorations, Materials 2020, 13, 5824; doi:10.3390/ma13245824
  7. Wołowiec-Korecka E., Czepułkowska-Pawlak W., Kula Z., Klimek L.: Effect of SiC Abrasive Blasting Parameters on the Quality of the Ceramic and Ni-Cr Dental Alloy Joint. Materials 2022, 15, 964. https://doi.org/10.3390/ma15030964
  8. Klimek L., Wołowiec-Korecka E., Czepułkowska-Pawlak W., Kula Z.: Quality of the Ceramic and Ni-Cr Alloy Joint after Al2O3 Abrasive Blasting, Materials 2023, 16, 3800. https://doi.org/10.3390/ma16103800

  1. Sandblasting process sometimes leads to hardening of the sample surface, which should be discussed in this manuscript. 

Answer: This was added in the article: " Although sandblasting hardens the surface of the metal, this has no practical significance, due to the hardening being negated by the recrystallization that occurs when the ceramic is fired at 900°C.”

Round 2

Reviewer 1 Report

The authors have addressed all my concerns and the paper can be accepted

The authors have addressed all my concerns and the paper can be accepted

Reviewer 2 Report

The authors made a series of improvements to the paper. However, I believe that the work can still be revised. The authors must inform themselves about the planning of the experiment and implement it in the present case.